# Diagnostic Imaging Features of Mammary Gland Tumors in Dogs and Cats

**DOI:** 10.3390/ani15243506

**Published:** 2025-12-05

**Authors:** Marisa Esteves-Monteiro, Joana Santos, Ana Patrícia Fontes-Sousa, Cláudia S. Baptista

**Affiliations:** 1Associated Laboratory for Green Chemistry of the Network of Chemistry and Technology (LAQV@REQUIMTE), University of Porto (UP), 4050-313 Porto, Portugal; 2School of Medicine and Biomedical Sciences, University of Porto (ICBAS-UP), 4050-313 Porto, Portugal; 3Veterinary Hospital of the University of Porto (UPVet), 4050-313 Porto, Portugal; 4Centre for Pharmacological Research and Drug Innovation (MedInUP/RISE-Health), Department of Immuno-Physiology and Pharmacology, ICBAS-UP, 4050-313 Porto, Portugal; 5Department of Veterinary Clinics, ICBAS-UP, 4050-313 Porto, Portugal; 6Associated Laboratory AL4AnimalS, Centre for the Study of Animal Science (CECA), University of Porto, 4099-002 Porto, Portugal

**Keywords:** canine, feline, mammary neoplasia, metastasis, diagnostic imaging techniques

## Abstract

Mammary gland tumors are among the most common cancers in female dogs and cats. This article explains how different imaging techniques, such as radiology, ultrasound, computed tomography (CT) scans, magnetic resonance imaging (MRI), and nuclear medicine, are used by veterinarians to detect, evaluate, and monitor these tumors. Imaging helps to identify whether a tumor is benign or malignant and to check if it has spread to other organs, like the lungs or lymph nodes. Although laboratory analysis of tissue remains essential for a final diagnosis, imaging plays a key role in planning treatment and follow-up care. Ultrasound is the most used method because it is simple, safe, and affordable, but more advanced techniques like CT or MRI provide extra detail when needed.

## 1. Introduction

The frequency of mammary gland tumors (MGT) varies significantly across species. However, they are the most common neoplasms in intact female dogs and represent a leading cause of mortality in this population [1]. Canine mammary tumors account for approximately 13.4% of all tumors diagnosed in dogs and 41.7% of those found in intact female [2]. The reported proportion of benign and malignant canine MGT varies considerably across studies. The classical literature often cites a distribution of approximately 60% benign and 40% malignant tumors [3]. However, more recent evidence indicates that this distribution is not uniform worldwide. According to a comprehensive review by Vazquez et al., 2023, the incidence and malignancy rates of mammary tumors exhibit marked geographic variation, largely influenced by regional differences in neutering practices, socioeconomic factors, and population management strategies [1]. For example, in certain regions, particularly where elective gonadectomy is less common, higher proportions of malignant tumors and increased metastatic potential have been documented [1]. Notably, in addition to the high proportion of benign tumors, many low-grade carcinomas also exhibit relatively benign behavior and do not metastasize [4]. The common practice of early-age ovariohysterectomy appears to significantly reduce the incidence of mammary neoplasia in female dogs, reinforcing the role of sex hormones as major risk factor [5,6]. However, recent evidence suggests that the effects of neutering may vary considerably among breeds. For instance, early spaying markedly reduced the risk of MGT in breeds such as the Dachshund and Yorkshire Terrier, while in others, including the Golden Retriever and Boxer, no clear protective effect was observed [7]. Additionally, male dogs are rarely affected, and when they are, the tumors are typically benign [8]. There also appears to exist a genetic predisposition, as certain breeds have been reported to have a higher risk of developing MGT. These include Brittany Spaniels, English Springer Spaniels, Labrador Retrievers, Irish Setters, Pointers, Great Pyrenees, Samoyeds, Airedale Terriers, Miniature and Toy Poodles, and Keeshonds [9]. The average age at diagnosis ranges from 10 to 11 years, and approximately 50% of affected bitches present with multiple tumors, which may exhibit distinct histological types [4].

Mammary tumors in cats are less common than in dogs [4] representing the third most common tumor in female cats after skin tumors and lymphoid cancers [10]. Although less prevalent, more than 85% of registered cases of neoplastic mammary changes in cats are malignant with adenocarcinomas being the most common histological type [11]. They occur at a mean age of 10–12 years, and like what happens in female dogs, ovariectomy also dramatically decreases the risk of developing mammary carcinoma [10,12]. MGT are reported to occur in various breeds of cats, but Siamese cats seem to have an increased risk of developing tumors at younger ages [9]. Cases in male cats are rare and represent approximately 1–5% of all feline mammary neoplasms [13].

All malignant MGT have the potential to metastasize. The risk and pattern of metastasis depend on the tumor type, histologic differentiation, and several clinical prognostic factors. In dogs, malignant epithelial tumors tend to metastasize via lymphatic routes to the regional lymph nodes and the lungs, whereas malignant mesenchymal tumors typically disseminate hematogenously, often resulting in direct pulmonary metastasis [1]. Dogs with mixed malignant MGT develop metastasis within the first year [2]. The lungs represent the most frequent site of metastasis, reported in approximately 81.4–83% of malignant mammary tumors in dogs [14,15]. Although less common, distant metastases have also been documented in several other organs, including the skin, liver, spleen, kidneys, adrenal glands, peritoneum, bones, brain, eyes, heart, urinary bladder, uterus, and small intestine [15,16,17,18,19,20,21,22]. In cats with mammary carcinomas, metastatic disease is common, typically involving the regional lymph nodes and/or the lungs, affecting more than 80% of cats [23,24]. These tumors often give rise to multiple pulmonary metastases, which can result in a diffuse interstitial pattern on radiographs, accompanied by pleural effusion [4]. In addition to the common pulmonary and nodal sites, feline mammary carcinomas may disseminate to more unusual locations such as bone, muscle, adrenal glands, and brain, underscoring the need for comprehensive evaluation in affected cats [25].

Histopathological examination remains the gold standard for accurately diagnosing mammary tumors [11]. In dogs with MGT, biopsy is recommended as the initial diagnostic approach, as fine-needle aspiration may not reliably distinguish between benign and malignant epithelial tumors [2]. Comprehensive clinical evaluation is also essential and should include a thorough physical examination, complete blood cell count, serum biochemistry profile, and urinalysis. If inflammatory mammary carcinoma is suspected, a coagulation profile is warranted due to its strong association with disseminated intravascular coagulation [9]. In human medicine, mammography and ultrasonography play a critical role in the diagnosis of mammary diseases; however, an ultrasound-guided biopsy remains essential for histopathological confirmation. While diagnostic imaging guidelines for mammary tumors are well established in human healthcare, comparable standardized protocols are lacking in Veterinary Medicine. To date, only an algorithm for the evaluation of sentinel lymph nodes in dogs with mammary carcinoma has been proposed [26]. In current veterinary clinical practice, radiology, ultrasonography (US), and computed tomography (CT) are routinely employed for the detection of mammary neoplasms and their metastasis in both dogs and cats. Additionally, Doppler US (utilizing both color flow and spectral modes) has shown utility in differentiating between benign and malignant mammary tumors [26].

Medical imaging techniques are fundamental tools in clinical oncology, providing essential information on disease extent and contributing significantly to treatment planning [27]. As Veterinary Medicine progressively adopts a wider range of imaging modalities commonly used in human medicine [26], this review aims to provide a comprehensive overview of the current state-of-the-art in imaging of mammary tumors and their metastases in veterinary patients. The objective is to concisely describe the available imaging techniques and illustrate their clinical relevance through practical examples, highlighting their diagnostic value in the management of this disease.

This article is a narrative (non-systematic) review. To ensure comprehensive coverage of the literature, a structured search was performed in PubMed, Scopus, and Web of Science using combinations of the terms “mammary gland tumor”, “canine”, “feline”, “diagnostic imaging”, “ultrasound”, “Doppler”, “CEUS”, “elastography”, “computed tomography”, “MRI”, and “nuclear medicine”. Studies published in English between 1980 and 2024 were considered. Inclusion criteria were: (i) studies involving dogs or cats with mammary tumors; (ii) studies evaluating imaging modalities; and (iii) original research articles, reviews, or case reports. Exclusion criteria included studies unrelated to imaging or not involving mammary tumors. Given the narrative nature of this review, no formal quality scoring system was applied, but emphasis was placed on studies with clear methodology and clinically relevant outcomes. Additionally, all figures in this review came from original imaging data obtained by the authors.

## 2. Imaging Techniques for the Diagnosis and Evaluation of Disease Progression

In canine and feline patients, the main imaging modalities used for the diagnosis, staging and monitoring of MGT include radiology (RX), various ultrasonographic techniques (e.g., B-mode, Doppler, contrast-enhanced ultrasonography (CEUS), three-dimensional (3D) US, and elastography), CT, magnetic resonance imaging (MRI), positron emission tomography (PET), and scintigraphy (Sci) [28].

The value of RX and CT lies primarily in their ability to detect pulmonary metastasis, which influences staging, prognosis, and therapeutic decision-making in affected animals [29,30]. Pulmonary metastatic lesions typically localize within the interstitial tissues rather than the alveolar or bronchial regions, often resulting in a lack of clinical signs or abnormalities on thoracic auscultation [30]. This asymptomatic nature underscores the critical role of imaging in identifying metastatic spread. Radiology remains the first-line modality for detecting pulmonary nodules, which represent the most common manifestation of metastases [30]. Standard three-view thoracic radiography (including right and left lateral views and either a dorsoventral or ventrodorsal view) is accepted as the baseline imaging protocol for pulmonary metastases screening [31,32]. However, thoracic X-ray has limited sensitivity, particularly in identifying small nodules, and tends to underestimate the prevalence of metastasis, particularly in dogs of larger breeds [31]. Studies suggest that its overall sensitivity for detecting pulmonary metastases of any size may be as low as 65%, with a minimum nodule diameter of approximately 4 mm required for radiographic visualization [14]. Despite its limitations, three-view thoracic radiography remains the standard imaging approach in routine veterinary practice due to its affordability, accessibility, and non-invasiveness compared to more advanced modalities [31].

CT is an advanced imaging modality used for the detailed assessment of anatomical structures, offering high spatial resolution and moderate tissue contrast differentiation [26]. In human medicine, CT is considered the gold standard for detecting pulmonary nodules, owing to its superior sensitivity in identifying small nodules compared to radiography [31]. Unlike in human patients, CT examinations in Veterinary Medicine are routinely performed under general anesthesia to minimize motion artifacts [26], that could compromise visualization of pulmonary parenchyma [33]. The evolution of CT in Veterinary Medicine has paralleled its use in human medicine, initially focusing on central nervous system evaluation in small animals, and expanding to encompass a wide range of indications, particularly oncologic staging [34]. Although primary lung tumors in dogs are relatively rare and usually affect elderly animals [35], pulmonary metastases, particularly from mammary gland carcinomas, are among the most prevalent secondary neoplastic conditions [29], reinforcing the importance of this technique. The two main advantages of CT, compared with thoracic radiography, are elimination of anatomical superimposition and superior contrast resolution. These capabilities enhance the detection of small pulmonary nodules that may be missed on radiographic exams [31].

Ultrasonography is a common imaging technique in veterinary practice, with abdominal US representing one of its most important applications in small animal clinical practice. This technique is commonly employed to assess parenchymal organs such as the liver, kidneys, urinary system, spleen, endocrine glands, lymph nodes, and gastrointestinal tract [36,37]. As previously mentioned, although less frequent than pulmonary metastases, metastatic involvement of abdominal organs has been reported in dogs and cats with mammary tumors [38,39]. Consequently, as with RX and CT, abdominal US is commonly included in the initial staging of bitches and queens with mammary tumors to detect potential metastatic spread to abdominal structures. Moreover, US can assist in the local evaluation of mammary lesions [38], including aiding in the prediction of malignancy based on sonographic features [40].

Several sonographic techniques, including advanced modalities, have shown promising utility in the evaluation of mammary tumors in companion animals [36]. B-mode (conventional US) remains the most widely used method and provides a non-invasive assessment of tissue architecture by processing grayscale echoes. This technique allows for the evaluation of mammary gland size, and parenchymal echotexture (homogeneous or heterogeneous) and echogenicity (hypo, hyperechoic, or mixed) [26,41]. Due to its accessibility and practicality, B-mode US remains the cornerstone for initial imaging of mammary lesions, offering valuable information about lesion characteristics [41].

Doppler US is a valuable tool for assessing blood flow within the mammary glands and provides important insights into the vascularization of tumoral tissues. This technique is typically performed after a B-mode scan and is integrated with it to generate duplex ultrasonography images. Color Doppler imaging uses a color scale to represent blood flow velocity and direction, facilitating the identification of vessel types [26]. However, one limitation of Doppler US is its reduced sensitivity in detecting small lesions or microcalcifications, potentially leading to false-negative results in the evaluation of mammary tumors in companion animals [42].

Elastography is an ultrasonographic technique designed to assess tissue elasticity and stiffness, thereby enhancing diagnostic sensitivity [43]. It has proven valuable in characterizing tumor lesions, with reduced tissue plasticity often suggesting malignancy [44]. Currently, two types of elastography techniques are employed: (1) Shear wave elastography provides quantitative insights into tissue stiffness by evaluating the speed at which shear waves, induced by acoustic radiation force, travel through the tissue; results are expressed in kilopascals or meters per second [45,46]; (2) Strain elastography is an imaging technique that provides real-time assessments of tissue elasticity by generating elasticity stiffness scores [47]. It is based on the principle that malignant tissues are generally stiffer than benign ones [48].

Contrast-enhanced US is an innovative imaging technique used both in human and Veterinary Medicine. It has demonstrated promising results in the evaluation of mammary tumors in humans [49,50,51], as well as in healthy ewes [52] and dogs [53]. More recently, CEUS has been applied to assess sentinel lymph nodes in dogs with mammary carcinoma [54]. The procedure requires specific software and the administration of a contrast agent. Video clips are captured immediately, and for a short duration, following contrast injection to document dynamic changes in vascularization. The benefits of CEUS examination encompass the visualization of neovascularization and the assessment of blood flow within functional vessels, making it a useful tool for evaluating mammary tumors. However, it has some limitations, including reduced sensitivity for detecting small lesions and relatively lower specificity compared to other imaging modalities [26].

Three-dimensional US is a relatively new imaging modality that has been used in human medicine [55], but its application in Veterinary Medicine remains limited, with usage reported in cattle [56]. In human medicine, 3D US has proven beneficial for guiding breast biopsies and surgeries involving breast neoplasia [55], as well as for distinguishing malignant from benign masses [57]. The potential utility of this technique in small animal practice has yet to be investigated.

PET scan is a radiotracer-based imaging technique that enables whole-body staging in a single imaging session [27]. It is a form of nuclear medicine that employs positron-emitting radionuclides attached to biologically important molecules involved in disease pathophysiology, serving as markers or participants [58]. This technique provides both qualitative and quantitative metabolic data, along with detailed anatomical visualization, particularly useful in tumor evaluation. One commonly used radiotracer in oncology is fluorine-18 fluorodeoxyglucose (^18^F-FDG), which capitalizes on the elevated glucose metabolism observed in most cancer cells due to their increased glycolytic activity [59]. The PET scan is commonly combined with CT to enhance diagnostic accuracy. CT provides detailed anatomical information about normal and pathological structures, while PET highlights areas of increased glucose metabolism. By fusing PET and CT images, healthcare professionals can more precisely determine whether regions of elevated metabolic activity correspond to anatomical abnormalities [60]. In Veterinary Medicine, PET scans have been tested in dogs for the detection of mammary tumors [27,61] and it is also useful for identifying metastasis, which typically exhibit hypermetabolic behavior and are readily visualized [60]. The advantage of PET lies in its ability to deliver true functional imaging [27]. Nevertheless, several limitations exist. When used without CT, PET lacks detailed anatomical information; additionally, the low sensitivity, and the high cost and limited availability of PET equipment in veterinary settings [60,62], along with the need for radioactive tracers, pose logistical challenges and potential health risks for both animals and operators [63].

Scintigraphy, like PET in the field of diagnostic nuclear medicine, involves the use of radiopharmaceuticals or radionuclides (radioactive isotopes or tracers) during imaging. These substances emit gamma radiation, allowing for the visualization and quantification of the distribution of various substances within the living organism. Unlike conventional imaging techniques, scintigraphy images do not provide anatomical detail but rather reflect functional activity, offering insights that other modalities may not capture [34]. This capability is particularly useful for evaluating tumor behavior and metastatic processes [64]. However, in Veterinary Medicine, scintigraphy is most used for diagnosing portosystemic shunts and is also an important diagnostic and staging tool for feline thyroid carcinoma cases [65,66]. The isotope Technetium-99m (^99m^Tc) is the most widely used radiotracer in both human and veterinary applications [64,67]. To date, only a single study has investigated the use of scintigraphy for detecting mammary tumor metastases in a cat, demonstrating its potential as a reliable method for identifying metastatic disease in feline mammary adenocarcinoma [68]. There are certain drawbacks associated with scintigraphy, including limited equipment availability, high operational costs, and the regulatory challenges of acquiring a radioactive materials license, which can be especially difficult in small animal practice [34]. Additionally, specialized training is essential for accurate interpretation of the images [65,69].

While not yet as widespread as CT, MRI is emerging as a significant modality in Veterinary Medicine. Its main advantage over CT lies in its ability to differentiate soft tissues structures, making it particularly valuable for evaluating tumors originating in soft tissue [58]. MRI operates by using a strong magnetic field to align the nuclear magnetization of hydrogen atoms, found in the body’s water content [26]. In human oncology, MR mammography is widely used, and in companion animals the technique is conducted following the principles established for human mammography [70]. However, to date, only one study has investigated the application of MRI in the evaluation of canine mammary tumors [70]. Additionally, MRI also has the potential to facilitate the study of lymph nodes that drain tumor regions in dogs [71]. Besides the excellent soft tissue contrast, MRI offers high spatial resolution (1 mm), all without exposing the patient to ionizing radiation or radioactive tracers [26]. Despite these advantages, several limitations hinder its broader use in veterinary practice. The high initial cost of MRI equipment and its ongoing maintenance requirements present significant financial barriers [58], contributing to its limited availability [26]. Another significant drawback of MRI is the length of the examination. Veterinary MRI scans can vary in duration depending on the number of sequences required and the anatomical regions under investigation. Furthermore, animals must be placed under general anesthesia during the procedure to prevent motion artifacts [34].

A summary of the diagnostic imaging techniques used for the evaluation and staging of MGT in dogs and cats is presented in Table 1.

## 3. Imaging Features of Primary Lesions

### 3.1. Ultrasonography

The sonographic parameters used to evaluate mammary tumors in companion animals include tumor size, width-to-length ratio, shape, margins, limits, nodular/mass echogenicity relative to surrounding tissues, echotexture, presence or absence of a hyperechoic halo, distal acoustic enhancement or shadowing, and signs of invasiveness [40,75,76].

Regarding tumor size, Nyman et al. reported a mean length of 19 mm (range: 3.9 to 60 mm) and mean width of 11 mm (range: 2.6 to 48 mm). A significant difference in both length and width was observed between benign and malignant tumors, with malignant tumors being larger. Also, benign mammary tumors were more frequently uniformly isoechoic or hypoechoic (Figure 1), whereas malignant tumors exhibited more variable echogenicity [75].

The width-to-length ratio also varies depending on the type and grade of mammary tumor [77]. Gonzalez et al. reported that all the malignant tumors in their study exhibited irregular margins, polymorphic shape, and heterogeneous internal echogenicity [76]. In fact, Bastan et al. identified spiculated or microlobulated margins as strong predictors of malignancy, with 76% of tumors displaying these features confirmed as malignant. An irregular shape was also highly indicative of malignancy, observed in 78% of malignant cases. Furthermore, they concluded that tumors with heterogeneous echogenicity are more likely to be malignant [78] (Figure 2).

Distal acoustic enhancement has been reported in a greater proportion of malignant tumors (22 out 30) compared to benign tumors (5 out 11) [75]. Marquardt et al. also found a correlation between acoustic enhancement and shadowing with malignancy [79]. However, Bastan et al. suggested that tumors displaying shadowing and/or enhancement are more likely to be benign [78], and other study reported that acoustic enhancement was observed equally in both benign and malignant tumors [80]. Similarly, distal acoustic shadowing was seen with equal proportions in both tumor types [75,76]. Taken together, the inconsistent associations reported in previous studies linking acoustic enhancement and shadowing variously with malignancy [79], benignity [81], or with no difference between tumor types [75,80], indicate that these features cannot be considered reliable standalone criteria for differentiating benign from malignant mammary tumors. Malignant neoplasia has also been associated with invasive behavior and poorly defined borders [76,82] (Figure 2), whereas benign tumors tend to be isolated [76] (Figure 1).

In a study involving 300 canine mammary masses, B-mode US demonstrated a sensitivity of 67.9% and a specificity of 67.6% for predicting malignancy [40]. The authors attributed the limited diagnostic performance of B-mode to the histopathological variability among tumor types, as well as the overlapping morphophysiological and structural characteristics of benign and malignant lesions. As a result, relying solely on B-mode ultrasonographic parameters becomes challenging for accurately differentiating mammary tumors and predicting malignancy [40].

On B-mode US, mammary gland hyperplastic lesions usually appear as small, well-circumscribed, homogeneous, and isoechoic to mildly hypoechoic areas relative to surrounding parenchyma, often associated with normal skin thickness and without evidence of invasiveness [40]. Benign mammary tumors, such as adenomas or mixed tumors, typically present as round to oval, well-defined nodules with regular margins and uniform or slightly heterogeneous echotexture [76]. In contrast, malignant mammary tumors are more often irregular in shape, with ill-defined or spiculated borders, heterogeneous echotexture [40,76,78], and possible acoustic shadowing or enhancement due to necrotic and fibrotic areas [40,75,76,79].

Elastography has been applied to the evaluation of MGT in both dogs [40,44,83,84,85,86] and cats [87]. In a study by Feliciano et al., malignant masses exhibited lower tissue deformability, with reported sensitivity, specificity, and accuracy of 75.6%, 66.7%, and 74.5%, respectively. Additionally, quantitative elastography demonstrated a high diagnostic performance in predicting malignancy in canine mammary masses, with sensitivity of 94.7%, specificity of 97.2%, and accuracy of 95.0% [40]. The same author also employed acoustic radiation force impulse (ARFI) imaging, finding it moderately accurate in identifying complex carcinoma types and potentially valuable for differentiating between benign and malignant mammary neoplasms [85]. In another study, Glińska-Suchocka et al. utilized shear-wave elastography to assess neoplastic mammary glands in dogs, revealing that benign neoplasms exhibited low stiffness, while malignant neoplasms were characterized by elevated stiffness [83]. Strain elastography also showed high accuracy in identifying intra- and peritumoral fibrosis through semi-quantitative analysis. However, its ability to differentiate among various carcinomas subtypes appear limited compared to more advanced elastographic techniques [44].

Elastography provides additional information about tissue stiffness. Hyperplastic and benign lesions demonstrate lower stiffness and greater deformability, whereas malignant tumors present significantly higher stiffness values, often exceeding 80–100 kPa on shear-wave elastography or high elasticity scores on strain imaging [44,83,87]. These differences reflect the denser stromal matrix and reduced elasticity typical of malignant infiltration. Quantitative elastography has shown diagnostic accuracies differentiating benign from malignant canine mammary masses [85].

Doppler imaging has proven useful in evaluating vascular patterns in canine mammary tumors. Vascular flow is commonly observed in both benign and malignant tumors, with no significant differences reported in several studies [75,81,88]. However, one study found that malignant tumors exhibited a higher degree of vascularization and an intermediate resistivity pattern, achieving a mean sensitivity of 86.0%, specificity of 47.9%, and an overall diagnostic accuracy of 81.5% for predicting malignancy in canine mammary masses [40]. Furthermore, increased neovascularization has been associated with malignancy [87]. Vascular distribution also differs by tumor type, as benign tumors typically present with a peripheral vascular pattern, whereas malignant lesions often demonstrate a mixed vascular pattern, characterized by an arbitrary vascularization both centrally and peripherally [74,80]. The reliability of ultrasonographic findings in assessing the malignancy of MGT in companion animals is closely associated with the echotexture characteristics observed in B-mode, specifically, the presence of homogeneous versus heterogeneous patterns, as well as in the distribution patterns of the vascular flow in Doppler studies. Although these imaging features offer valuable diagnostic insights, definitive confirmation still requires histopathological examination [80].

In Doppler mode, benign and hyperplastic lesions generally exhibit scant or peripheral vascularization with low-velocity flow, reflecting limited angiogenesis [75,80]. Malignant lesions tend to show increased vascular density, irregular or mixed (central and peripheral) vascular patterns, and higher resistive index and pulsatility index values, consistent with neoplastic neovascularization [67,74].

In addition to conventional Color Doppler, Power Doppler can be used to assess vascular patterns in MGT. Power Doppler detects the amplitude of the Doppler signal rather than the velocity or direction of blood flow, providing increased sensitivity for identifying low-velocity or small-caliber vessels that may not be visible with conventional Doppler. This makes it particularly useful for characterizing the intratumoral and peritumoral vascular architecture of mammary neoplasms. In canine mammary tumors, Power Doppler has been shown to reveal more extensive vascularization compared with Color Doppler, especially in malignant lesions, which tend to display chaotic or centrally located vascular signals, reflecting neoangiogenesis and invasive behavior [80].

CEUS uses microbubble contrast agents to assess macro- and microvascular perfusion and has shown increasing applicability in the evaluation of MGT in veterinary patients. In human patients, CEUS reveals distinct enhancement patterns between benign and malignant lesions. Malignant tumors, such as carcinomas, typically exhibit hyper-enhancement with a heterogeneous contrast distribution, while benign lesions often display hypo-enhancement with a homogeneous distribution of the contrast medium. One particularly specific enhancement pattern identified in malignant tumors is the presence of radial vessels, which demonstrate a high specificity of 97.7% [49]. In veterinary studies, CEUS has shown promise as a non-invasive tool for evaluating tissue perfusion (macro and microcirculation) in canine mammary tumors. It appears especially useful for detecting complex mammary carcinomas with moderate diagnostic accuracy. This diagnostic capability is attributed to the observed shorter periods of contrast wash-in and peak enhancement times in such tumors [77]. In a related study by the same author, a rapid washout time (shorter than 80.5 s) yielded a sensitivity of 80.2%, specificity of 16.7%, and accuracy of 77.4% [40]. These findings highlight CEUS as a useful adjunct tool for assessing tumoral perfusion patterns, despite its diagnostic limitations.

Despite its advantages namely real-time visualization of neovascularization and microvascular flow, CEUS has some limitations, including lower sensitivity for small lesions and modest specificity, as noted in previous comparative studies [26]. Nonetheless, the technique contributes meaningful functional information and may improve confidence in differentiating benign from malignant mammary lesions when used alongside B-mode, Doppler, and elastography.

### 3.2. Computed Tomography

To the authors’ knowledge, CT has been used in the diagnosis of canine mammary tumors in only one published study [72]. In that study, CT demonstrated a sensitivity, specificity, and accuracy of approximately 83.33% for identifying malignant tumors. Additionally, a significant correlation was observed between primary tumor size and malignancy. CT may offer valuable diagnostic benefits in cases of mammary tumors by providing detailed anatomical visualization and precise detection of calcifications [72]. However, further studies are needed to fully establish the clinical utility of CT in the diagnosis of mammary tumors in small animals.

Although published CT descriptions of canine and feline MGT remain scarce, certain imaging patterns can be inferred from the available data and from analogous findings in ultrasonography, histopathology, and human breast imaging. Benign tumors generally present as well-defined, homogeneous soft-tissue masses with smooth margins and mild or homogeneous contrast enhancement [72]. Malignant tumors, in contrast, tend to be larger and exhibit irregular or spiculated margins, heterogeneous internal attenuation due to necrosis or cystic degeneration, and heterogeneous or peripheral contrast enhancement [26] (Figure 3 and Figure 4). Areas of skin thickening or invasion of adjacent tissues may also be visible [26]. Mixed mammary tumors, which often contain cartilaginous or osseous metaplasia, can show foci of mineralization or calcified matrix within otherwise soft-tissue density lesions, findings that are easily recognized on CT images [89]. These features, although not pathognomonic, may provide useful clues to tumor behavior and guide sampling or surgical planning. Nevertheless, systematic CT–histopathological correlation studies are still lacking, and further research is needed to define objective criteria for differentiating benign, malignant, and mixed mammary tumors in small animals.

### 3.3. Positron Emission Tomography

Recent studies have demonstrated the potential of 18F-2-deoxy-2-fluoro-d-glucose PET/CT (FDG-PET/CT) in veterinary oncology for improving tumor staging and treatment planning [90]. In dogs with various malignant tumors, FDG-PET/CT provided additional diagnostic information that significantly influenced therapeutic decisions and prognosis. Similarly, in the case of canine inflammatory mammary carcinoma, FDG-PET/CT enabled the detection of extensive local infiltration and metastatic lesions, including in clinically normal skin. These findings highlight the usefulness of FDG-PET/CT in identifying both primary and metastatic tumor activity, refining staging accuracy, and guiding appropriate clinical management in canine oncology [91]. In other study, researchers analyzed the maximum standardized uptake value of glucose to assess its correlation with tumor size and its ability to differentiate between benign and malignant lesions. The results indicated that a minimum tumor size of 1.5 cm was required to distinguish malignancies, as lesions below this threshold, regardless of pathology, did not show significant differences in glucose uptake. Notably, a glucose uptake value greater than 2 demonstrated 100% sensitivity for detecting malignancy. When both criteria were combined, lesion size over 1.5 cm and glucose uptake exceeding 2, a positive predictive value of 100% was achieved. However, no association was found between glucose uptake and the histological subtype or tumor grade [27]. In a study in dogs with spontaneously occurring tumors, a canine mammary papillary cystadenocarcinoma showed strong tracer uptake on PET imaging, accompanied by a relatively rapid washout, further supporting the potential diagnostic utility of PET in characterizing mammary tumors [61].

### 3.4. Magnetic Resonance Imaging

The only study to date utilizing MRI for the evaluation of canine mammary tumors demonstrated that static MRI is effective in assessing key morphological features, including tumor size, shape, and tissue structure [70]. The tumors examined exhibited heterogeneous structure, some containing fluid content, and were clearly distinguishable from abdominal musculature, showing possible adherence to the skin or fascia. The study concluded that MRI allows for accurate localization and size measurement of tumors, providing valuable preoperative information for surgical planning or biopsy. Moreover, coronal and transverse scans performed with the animal in dorsal recumbency yielded the most diagnostically useful images [70]. Dynamic contrast-enhanced MRI further provided insights into tumor physiology, particularly microcirculation, with most tumors displaying early enhancement followed by a washout phase [70].

## 4. Imaging Features of Metastatic Lesions

### 4.1. Radiography

X-ray imaging is primary used to detect pulmonary metastases, which are the most common form of distant metastases associated with MGT in small animals [92]. Radiographic presentations of lung metastases can vary and may include well-defined nodules, poorly demarcated nodules, or pleural effusion in the absence of visible pulmonary lesions [93] (Figure 5 and Figure 6). In a study involving 136 dogs with MGT, 122 cases (89.7%) showed no evidence of intrathoracic metastatic disease. Among those with detectable metastases, pulmonary nodules were the most frequently observed radiographic finding, followed by peribronchial opacities and enlargement of the sternal lymph nodes. Of the six dogs that underwent necropsy, five were confirmed to have pulmonary metastases, but only one exhibited radiographic evidence of intrathoracic metastatic disease. Based on these findings, the authors concluded that radiography lacks sensitivity for detecting early intrathoracic metastases associated with MGT [14]. This observation is consistent with another study involving 21 dogs with mammary tumors, in which no pulmonary metastases were identified on radiographs, yet CT scans revealed metastatic lesions in the lungs [93]. Another study reported atypical radiographic patterns, such as linear interstitial opacities, that were confirmed as metastases originating from mammary adenocarcinoma. However, due to the limited number of cases included, the authors were unable to determine the prevalence of pulmonary metastatic lesions associated with mammary tumors [30].

A more recent study provided a broader characterization of pulmonary metastatic patterns, identifying pulmonary nodules as the most common finding, followed by pulmonary micronodules, miliary nodules, and pulmonary masses (Figure 5). These lesions exhibited a positive correlation with the malignancy of the primary mammary tumor. Additional radiographic patterns included interstitial disseminated reticulonodular and mixed disseminated alveolar interstitial presentations. In advanced stages of metastasis, atypical features such as calcifications, secondary pneumothorax, and cavitary signs were also observed. The study concluded that thoracic radiography proved to be a reliable, cost-effective, and time-efficient toll for detecting both early and advanced stages of pulmonary metastasis in cases of mammary neoplasia [94]. The authors suggested that the higher sensitivity observed in their findings may be attributed to recent advancements in radiographic imaging technology. Interestingly, in cats, pulmonary metastases from mammary tumors tend to present more frequently with atypical radiographic patterns, such as ill-defined nodules or diffuse pattern [95].

Radiographic evaluation of uncommon sites of distant metastases from canine and feline MGT has been poorly documented. Baptista et al. reported that the radiographic features of metastatic bone lesions were characterized primarily by moth-eaten lysis, accompanied by areas of new bone formation that were moderately mineralized and exhibited irregular margins [15] (Figure 7).

### 4.2. Computed Tomography

Thoracic CT is considered more sensitive than radiography for detecting pulmonary metastases. In a study involving 21 female dogs with mammary tumors, CT examinations revealed no abnormalities in the chest wall or mediastinum; however, lung metastases were identified in two cases. Additionally, small, focal atelectatic areas were detected in 28.57% of the dogs, typically presenting as peripheral infiltrates. The metastatic lesions appeared as solid nodules with well-defined margins, ranging in size from 0.4 and 0.7 cm (Figure 8). These findings underscore the superior sensitivity of CT compared to plain radiography in detecting pulmonary metastases [93]. These results were corroborated by another study of 20 female dogs with mammary tumors. In this study, thoracic CT identified subtle alterations such as discrepancies between bronchial and vascular diameters, thickening of the interlobular septa, increased visualization of centrilobular points, and small pulmonary nodules measuring up to 5 mm in 25% of the cases. Notably, these lesions were not visible on radiographs [96]. Also, in large breed dogs, thoracic and abdominal CT should be considered, since it has been demonstrated that in dogs weighing more than 25 Kg, CT detected significantly more lesions than ultrasound, including lesions considered clinically relevant [97].

Similar findings have been reported in feline patients. A study in cats also demonstrated that CT could detect smaller pulmonary nodules than those visible on thoracic radiographs, with the most common nodule location being the caudal thoracic region. The study detected intra-thoracic lymphadenopathy and pleural effusion. However, it was noted that general anesthesia can induce lung atelectasis, potentially affecting the accuracy of pulmonary assessments. Atelectatic regions may display interstitial or alveolar patterns, which can obscure pathological lung lesions, complicating diagnostic interpretation [33].

In studies investigating sentinel lymph node metastasis in canine mammary tumors, contrast-enhanced CT demonstrated high sensitivity and specificity (87.5% and 89.3%, respectively) (Figure 9 shows an example in a cat). Notably, in 88% of metastatic lymph nodes, peak contrast enhancement occurred within the first 3 min following contrast medium administration, likely reflecting increased vascularization associated with neoplastic infiltration [28]. Another study by the same authors found that a heterogeneous enhancement pattern or absence of opacification, accompanied by reduced enhancement values observed one-minute post-contrast medium injection, was strongly associated with neoplastic invasion of sentinel lymph nodes. In contrast, lymph nodes size and shape were found to be less reliable criteria. Most metastatic nodes displayed heterogeneous patterns, characterized by spotted, partial, or peripheral contrast opacification, or the absence of opacification. Based on these findings, the authors concluded that the post-contrast opacification pattern is a useful predictor of sentinel lymph node metastasis in dogs with mammary neoplasia [73].

### 4.3. Ultrasound

Various ultrasound techniques are frequently used to evaluate sentinel lymph nodes in dogs with mammary tumors, a critical step in determining prognosis. Metastases primarily occur in the inguinal and axillary locoregional lymph nodes and are associated with an unfavorable prognosis [36].

In a study by Muramoto et al., B-mode US was used to assess lymph node characteristics, including size, margins, shape, architecture, echotexture, and echogenicity. Doppler US further evaluated the vascular pattern by analyzing the quantity and distribution of blood flow. Ultrasonographic findings were compared to histopathologic results. Lymph nodes were classified as metastatic on B-mode imaging based on the presence of necrotic, liquefactive, or hemorrhagic areas, normal and tumor-altered tissues, and evidence of micro or macro calcifications, resulting in a heterogeneous echotexture. The ultrasonographic classification of the lymph nodes was accurate in 92.5% of cases, with a sensitivity of 94.1%, and a specificity of 92% [98]. Silva et al. examined 196 lymph nodes (100 inguinal and 96 axillary) in B-mode from 100 female dogs with mammary tumors. They found that metastatic lymph nodes exhibited a decreased short/long (S/L) axis ratio, with an overall diagnostic accuracy of approximately 60% in terms of identifying altered nodes.

These findings were consistent with those of Stan et al., who reported that an S/L axis ratio greater than 0.55 was the most significant predictive sign for metastases in sentinel lymph nodes, achieving a sensitivity of 83.3% and a specificity of 78.6%. Additionally, some metastatic nodes showed inhomogeneous echostructure with evidence of coagulative necrosis [54]. A similar study in feline patients also showed that metastatic lymph nodes exhibited alterations in size and shape, with some displaying a well-defined central hyperechoic area [99].

In Doppler ultrasound examinations, vascular evaluation using color Doppler has shown variable diagnostic utility for detecting lymph node metastasis. In the study by Silva et al. [45], color Doppler imaging did not provide reliable parameters for identifying metastatic involvement. However, Moraes et al. showed that metastatic lymph nodes presented peripheral neovascularization or a mixed pattern [99]. These findings were further supported by Stan et al., who reported that 88.88% of metastatic sentinel lymph nodes displayed disorganized, chaotic vascularization, with predominantly peripheral (55.55%) or mixed (20.37%) blood flow patterns. Additionally, the study demonstrated that hemodynamic indices obtained through pulsed-wave Doppler could distinguish between benign and malignant processes in affected lymph nodes. Specifically, a resistive index (RI) greater than 0.56 and a pulsatility index exceeding 1.02 were indicative of metastatic infiltration [54]. Based on these results, the authors concluded that the S/L ratio and RI are the most reliable sonographic parameters for distinguishing between normal and metastatic sentinel lymph nodes when using a combination of B-mode and Doppler ultrasound [54].

Acoustic radiation force impulse elastography showed that metastatic lymph nodes had less deformability compared to normal nodes. Quantitative analysis revealed that shear wave velocity was highest in metastatic nodes, followed by reactive and normal nodes. Shear wave velocity enabled the accurate identification of metastasis in both inguinal (sensitivity 95%, specificity 87%) and axillary lymph nodes (sensitivity 100%, specificity 94%) [45]. These findings were supported by another study, which also reported significantly higher stiffness in metastatic lymph nodes compared to non-affected ones. This difference in elasticity resulted in high diagnostic accuracy when using elasticity scoring systems to detect metastatic involvement [54].

The CEUS technique has been evaluated in only one study. The study found that most unaffected sentinel lymph nodes (76.19%) display enhancement patterns without defects, while metastatic nodes typically exhibited an inhomogeneous pattern (81.48%) with areas lacking contrast uptake. While mean wash-in times were similar between metastatic and unaffected nodes, wash-out times differed significantly. Metastatic nodes demonstrated a shorter and more pronounced wash-out phase, suggesting the more rapid clearance of contrast agent. These findings indicate that wash-out time may serve as a more reliable predictor of metastatic involvement than wash-in characteristics [54].

As previously mentioned, abdominal US is routinely included in the initial clinical evaluation of patients with mammary tumors. While its diagnostic and prognostic value has been extensively studied in canine neoplasms such as osteosarcoma and mast cell tumors, there is only one major study addressing its role in mammary tumors. In this study, 201 female dogs with mammary tumors were evaluated, and 78% presented with at least one abnormal abdominal finding in US. The spleen was the most frequently affected organ, followed by the uterus and liver. Importantly, dogs with regional lymph node metastasis were significantly more likely to exhibit abdominal abnormalities. The most common alterations indicative of metastasis included heterogeneous hypoechoic nodules in the spleen, focal heterogeneous hepatic masses with cystic areas, nodules in the adrenal glands and kidneys, and enlarged and heterogeneous iliac and mesenteric lymph nodes [38].

Baptista et al. described the ultrasonographic appearance of a metastatic ocular mammary gland carcinoma as a large, irregular, and heterogeneous mass. The A-scan echogram revealed prominent acoustic interfaces with high internal reflectivity, reduced sound attenuation, irregular spikes, and limited vascularization [21].

### 4.4. Nuclear Medicine

Technetium-labeled scintigraphy is widely used for staging neoplasms such as osteosarcomas and lymphomas in small animals; however, its application in mammary tumors is scarce. In a case study involving a cat with mammary tumors, ^99m^Tc-THY scintigraphy revealed increased focal uptake in the liver and lungs, along with notably high radionuclide accumulation in the right kidney. This renal uptake was subsequently confirmed as metastasis, lesions that had not been detected by radiography or US. Consequently, the authors recommended the use of ^99m^Tc-THY scintigraphy as a complementary tool for the staging of mammary tumors in veterinary practice [68]. These observations were further confirmed by another case involving a female dog with mammary tumors. In this instance, pulmonary metastasis was diagnosed via ^99m^Tc-THY scintigraphy, before it became evident on thoracic radiographs. The scintigraphic findings were later confirmed as metastatic lesions upon necropsy, emphasizing the superior sensitivity of scintigraphy for early detection of metastatic spread compared to conventional imaging [100].

It is important to note that the available literature presents significant heterogeneity in study design, sample size, imaging protocols, equipment settings, and histopathological classifications. These factors limit direct comparison between studies and may partially explain the variability in reported diagnostic accuracies across imaging modalities. As a narrative review, the present synthesis aims to highlight consistent trends, while acknowledging that standardized imaging protocols and larger, controlled studies are still needed to strengthen the evidence base.

Species-specific considerations are also relevant when interpreting imaging findings. Feline mammary tumors are biologically more aggressive, with higher malignancy rates and faster metastatic progression compared with canine tumors, which may influence the clinical interpretation of certain imaging characteristics [1]. Furthermore, most published studies evaluating Doppler, CEUS, elastography, CT, and MRI include predominantly canine populations, with feline cases often underrepresented or absent [75,77,80]. Therefore, the diagnostic reliability and predictive value of several imaging criteria established for dogs cannot be directly extrapolated to cats, underscoring the need for species-specific evidence and cautious interpretation.

## 5. Final Considerations

The aim of this review was to evaluate the diagnostic usefulness, strengths, and limitations of the main imaging modalities used in the assessment of MGT in dogs and cats. Overall, imaging contributes to essential morphological and functional information for characterizing primary tumors, assessing regional and distant metastases, guiding sampling, supporting surgical planning, and establishing follow-up strategies. While histopathology remains the gold standard due to overlapping sonographic and cross-sectional features between benign and malignant lesions, imaging provides critical clinical value throughout the diagnostic and staging pathway. However, current evidence is limited by heterogeneous study designs, variable imaging protocols, differences in equipment and operator experience, and the predominance of canine-based studies, factors which restrict direct extrapolation to feline patients.

For primary tumor and regional lymph node assessment, US is currently recommended since it is a readily available, non-invasive, and radiation-free technique that allows the local evaluation of tumors and provides real-time guidance for sampling procedures. Advanced sonographic techniques, including Doppler, CEUS, and elastography, can further aid in differentiating between benign and malignant lesions by assessing vascular and mechanical properties of the tissue. CT should also be considered for the evaluation of regional lymph nodes and distant metastases, as it offers high spatial resolution and the ability to detect small pulmonary or abdominal lesions that may be missed by radiography or ultrasound. Contrast-enhanced CT has also proven useful for the identification of metastatic sentinel lymph nodes, based on enhancement patterns rather than size criteria.

Malignant imaging features of primary tumors are mainly characterized by (i) irregular or lobulated shape (vs. round/oval); (ii) ill-defined or spiculated margins (vs. well-circumscribed); (iii) increased central or anarchic flow on Doppler US (vs. peripheral flow); (iv) skin thickening or local invasion; and (v) increased tissue stiffness on elastography, reflecting reduced elasticity commonly associated with malignant lesions (Table 2).

Common sites of MGT metastasis include the lungs and regional lymph nodes, while infrequent locations include the liver, spleen, bones, kidney, adrenal glands, heart, CNS, eye, muscle, peritoneum, skin, urinary bladder, uterus or small intestine. For staging and follow-up purposes, three-view thoracic radiographs—or thoracic CT when available—together with abdominal US are recommended. If CNS clinical signs are present, MRI is the most appropriate modality to evaluate intracranial metastatic disease. Thoracic metastases most commonly present as multiple, well-defined nodules, although atypical patterns such as miliary, reticulonodular, or alveolar opacities may occur, particularly in late-stage disease. Pulmonary CT typically reveals small nodules (3–8 mm) that may not be visible on radiographs, reinforcing its superior sensitivity.

Although physiological uptake can complicate interpretation, nuclear medicine techniques such as scintigraphy and PET provide functional whole-body information that may reveal micrometastases. FDG-PET/CT has demonstrated the ability to detect both primary and metastatic lesions, improving staging accuracy and therapeutic planning. However, these techniques remain limited by high cost, restricted availability, and the need for specialized facilities.

To support practical clinical decision-making, imaging modalities should be selected according to the diagnostic objective and the stage of disease. For initial evaluation of the primary mammary mass and regional lymph nodes, conventional B-mode US remains the recommended first-line tool, given its accessibility, safety, and ability to guide sampling procedures. When additional information regarding vascularity or tissue stiffness is required, particularly in cases where malignancy is suspected, Doppler, CEUS, and elastography can be incorporated to refine diagnostic confidence.

For staging purposes, thoracic radiography continues to be the baseline method to screen for pulmonary metastases; however, when available or when radiographic findings are equivocal, thoracic CT should be preferred due to its markedly superior sensitivity for detecting small nodules. Abdominal US is advised for routine evaluation of possible abdominal metastatic spread, while MRI should be reserved for patients presenting with neurological signs or when soft-tissue contrast resolution is crucial for treatment planning. Nuclear medicine techniques, including scintigraphy and PET, may be considered as adjunct tools in selected cases where functional information or whole-body screening is required, recognizing their current limitations in terms of cost, availability, and need for specialized equipment.

Future developments in advanced imaging techniques are expected to further enhance diagnostic accuracy in canine and feline MGT. Quantitative CEUS may improve the assessment of tumor perfusion and microvascular characteristics, enabling more robust discrimination between benign and malignant lesions. The application of radiomics and texture analysis to CT, MRI, or US datasets holds promise for extracting high-dimensional imaging biomarkers linked to tumor phenotype and prognosis. Additionally, the integration of artificial intelligence and machine learning algorithms may support automated lesion detection, segmentation, and risk stratification, ultimately improving staging consistency and reducing operator-dependent variability. The standardization of acquisition protocols and the validation of multimodal imaging biomarkers represent important future research priorities to advance the clinical utility of diagnostic imaging in veterinary mammary oncology.

In conclusion, multimodal imaging integrating morphological and functional techniques provides the most comprehensive and clinically relevant approach for the diagnosis, characterization, and staging of MGT.

## Figures and Tables

**Figure 1 animals-15-03506-f001:**
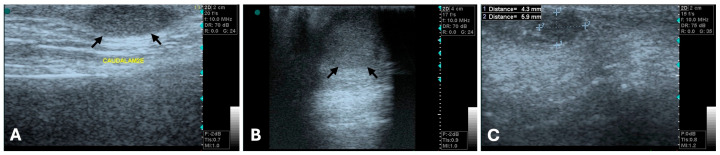
B-mode ultrasonograms, long-axis images (General Electric Vivid 3, 10 MHz linear probe). Benign mammary tumors (between arrows or clippers) with round to oval shape, well-defined regular margins, and hypoechoic, homogenous echotexture, features typically associated with benign lesions. (**A**) Mixed benign tumor, (**B**) complex adenoma, and (**C**) simple adenoma.

**Figure 2 animals-15-03506-f002:**
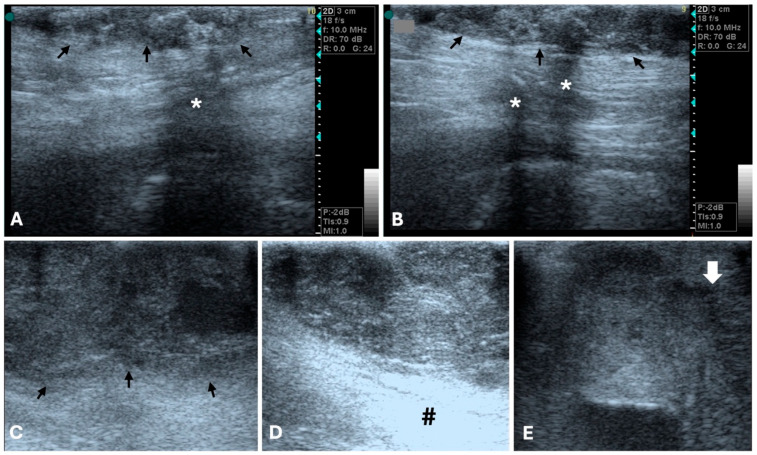
B-mode ultrasonographic appearance of canine malignant mammary tumors in longitudinal plan (General Electric Vivid 3, 10 MHz linear probe). (**A**,**B**) Grade I mammary tubular carcinoma (arrows), with irregular, ill-defined margins, and mixed, heterogenous echotexture. Acoustic shadowing is evidenced (*). (**C**,**D**) Tubular carcinoma with vascular invasion, showing lobulated margins (arrows), and mixed, heterogenous echotexture. Distal acoustic enhancement is evidenced (#). (**E**) Grade III solid carcinoma. Note the heterogeneous echogenicity with multiple irregular hypoechoic areas of variable sizes. Tumor limits are poorly defined, and margins are irregular and spiculated (arrow is evidencing the tumor border with the previous characteristics).

**Figure 3 animals-15-03506-f003:**
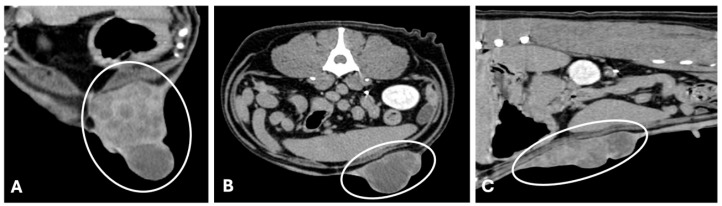
Computed tomography (General Electric Lightspeed, 16 slices scanner) of a canine mammary tubular carcinoma with vascular invasion (late venous/excretory phase). (**A**,**B**) axial image, and (**C**) sagittal reconstruction. Large mass with 18 cm length × 5.7 cm width × 7 cm thickness (white circumferences). The mass shows irregular lobulated contours and ill-defined interfaces with adjacent musculature, indicating infiltrative behavior. In the venous phase, the mass evidenced heterogeneous contrast enhancement with central hypoattenuating foci of necrosis (histologically confirmed), a feature typical of high-grade malignancies.

**Figure 4 animals-15-03506-f004:**
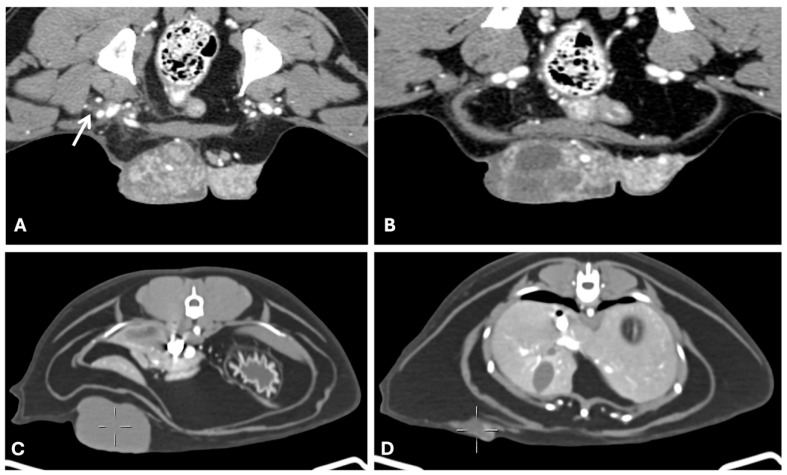
Computed tomography of canine and feline malignant mammary gland tumors ((**A**,**B**) General Electric Lightspeed, 16 slices scanner; (**C**,**D**) Siemens Somaton go. UP 32 slices), axial images. (**A**,**B**) Grade III canine mammary solid carcinoma with vascular invasion (venous phase). Large lobulated mass with heterogeneous contrast uptake due to hypodense areas of necrosis (histologically confirmed). The mass had random internal vascularization and was poorly demarcated to the right inguinal adipose tissue with fat stranding from the tumor surface towards the inguinal ring (arrow). (**C**,**D**) grade II intraductal feline papillary carcinoma. Well defined multiple masses along the right chain of a female cat, with irregular lobulated contour and homogenous hypodense density (cystic nature). Tumors varied from 4.5 cm length × 4.32 cm width × 2.3 cm thickness to small irregular small infracentimetric cystic masses. Masses were circumscribed to the adipose tissues.

**Figure 5 animals-15-03506-f005:**
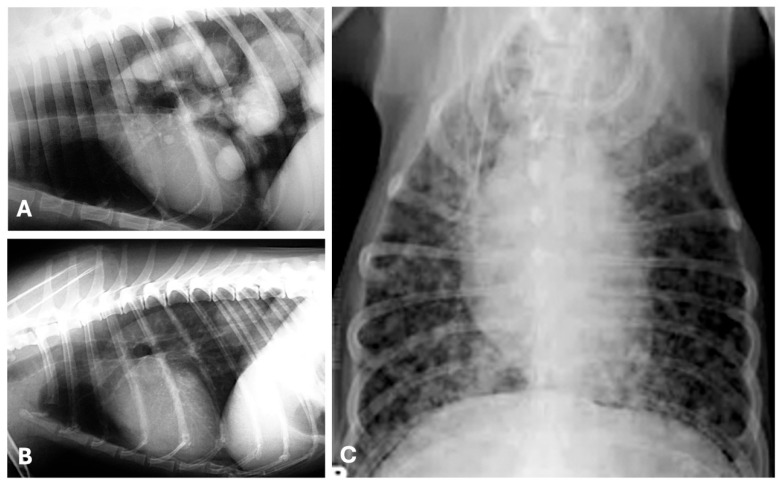
Thoracic radiographs showing pulmonary metastases from canine mammary tumors. Right (**A**) and left (**B**) lateral, and ventro-dorsal (**C**) radiographic views illustrating metastatic disease in the lungs from a tubulopapillary (**A**,**B**) and a malignant mixed mammary tumor (**C**) in three canine patients. (**A**) Multiple well-circumscribed, soft tissue density nodules/masses (“cannonball” pattern), mainly localized in the caudal, medial and accessory lung lobes (predominant caudo-dorsal distribution). (**B**) Single soft tissue mass, well-defined, localized in the left caudal lung lobe. (**C**) Miliary/micronodular, ill-defined soft tissue nodules scattered throughout the lung field.

**Figure 6 animals-15-03506-f006:**
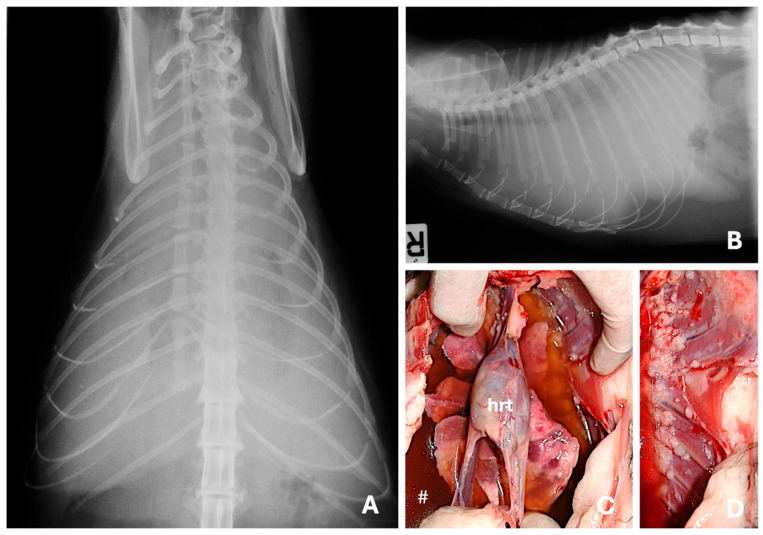
Ventro-dorsal (**A**) and right lateral (**B**) radiographic views evidencing severe pleural effusion in a domestic shorthair cat with a mammary solid carcinoma. Besides the pleural fluid (#), at the necropsy were observed multiple, metastatic small nodules spread throughout the lungs (**C**) and pleura (**D**). Pleural effusion is a common presentation of metastatic feline mammary carcinoma and indicates advanced disease. (hrt) Heart.

**Figure 7 animals-15-03506-f007:**
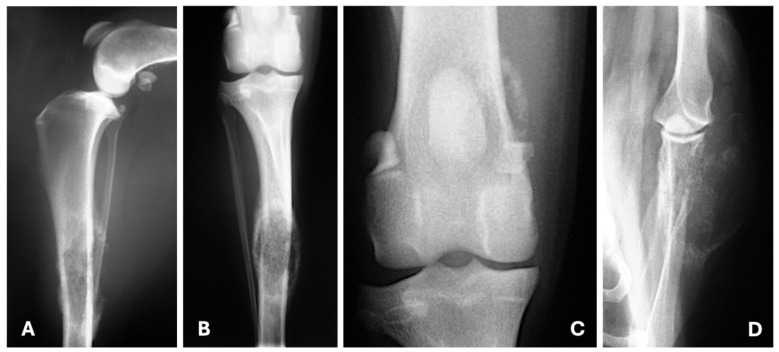
Medio-lateral (**A**), craniocaudal (**B**,**C**) and caudocranial (**D**) radiographic views illustrating bone metastasis from a tubulopapillary ((**A**–**C**)—polyostotic disease) and a solid mammary carcinoma (**D**). (**A**,**B**) Moderate soft tissue swelling surrounds the middle third of the diaphysis of the tibia; in this area is also visible moth-eaten lysis and new bone formation moderately mineralized and irregularly marginated. (**C**) New bone formation moderately mineralized in the medial aspect of the distal femur. (**D**) Moth-eaten lysis in the distal scapula with new bone formation predominantly in its lateral aspect and soft tissue swelling.

**Figure 8 animals-15-03506-f008:**
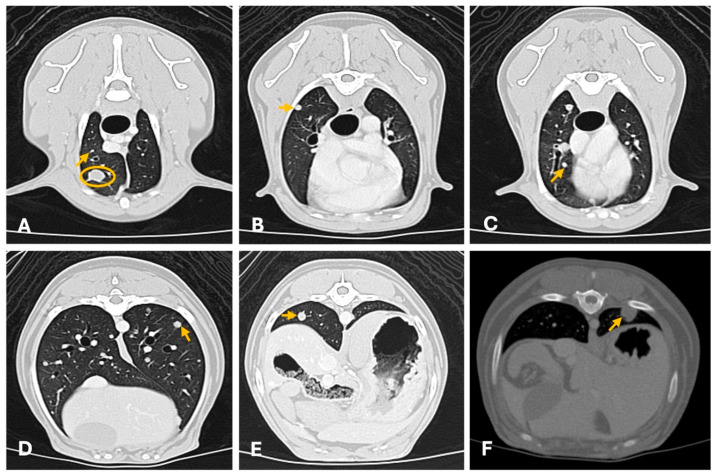
Thoracic computed tomography (General Electric Lightspeed, 16 slices scanner) showing multiple (**A**–**E**) and single (**F**) pulmonary metastasis of canine mammary tumors (axial images). (**A**–**E**) Carcinosarcoma (lung window ww 1500, L -500; 1.3 mm thickness). There are multiple widespread well-defined small pulmonary nodules of different sizes (from 3 mm to equal to 8 mm; arrows evidence a few examples) and a well-defined 1 cm nodule in the ventral cranial right lung lobe (yellow circumference). All nodules were homogeneous with moderate contrast uptake. (**F**) Tubular carcinoma. Well defined solitary subpleural solid homogeneous nodule (1 cm × 0.8 cm diameter) in the left dorso-caudal lung lobe with moderate homogeneous contrast uptake (arrow). CT provides superior sensitivity for detecting small metastatic nodules not visible on radiography.

**Figure 9 animals-15-03506-f009:**
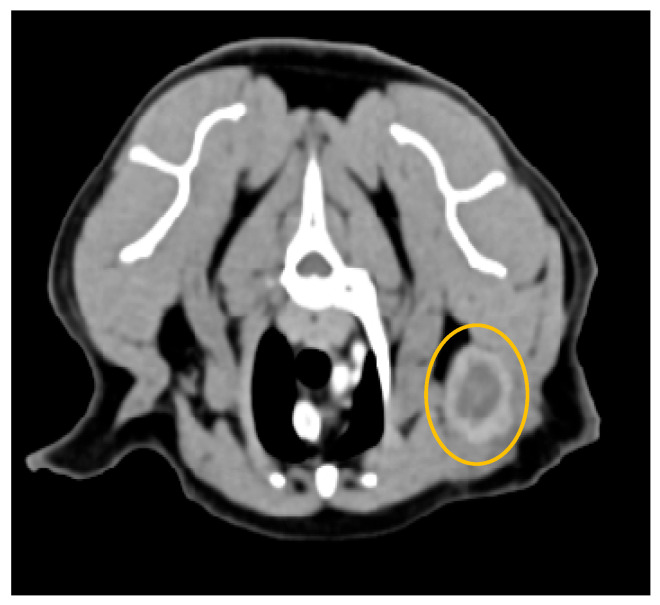
Computed tomography of a metastatic left axillary lymph node (yellow circumference), from a feline mammary tubulopapillary carcinoma (General Electric Lightspeed, 16 slices scanner; axial image). Intense peripheral ring contrast enhancement and irregular lobulated contour with a hypodense center of neoplastic epithelial malignant cells. The lymph node measured 3 cm × 0.5 cm. Heterogeneous or peripheral enhancement patterns strongly correlate with metastatic infiltration.

**Table 1 animals-15-03506-t001:** Diagnostic imaging techniques used for the evaluation and staging of mammary gland tumors in the dog and cat.

Imaging Technique	Advantages	Disadvantages	Primary Mass	Metastases
RX	Quick, highly available, low cost [37]	Low sensitivity, superimposition of structures, low differentiation of soft tissues [37]	No [72]	1st line (pulmonary metastases) [30]
CT	Removal of superimposition of structures, high sensitivity, contrast resolution and spatial resolution [34]	Low soft tissue contrast, cost, low availability, need for anesthesia [34]	Yes [72]	2nd line (pulmonary, lymphatic and abdominal metastases) [31,73]
B-mode US	Quick, highly available, low cost, safe [37]	Requires proper education of the operator [37]	Yes (size, location, echogenicity) [74]	1st line (abdominal metastases) [37]
Doppler US	Quick, highly available, low cost, safe [37]	Inability to detect small lesions or microcalcifications [75]	Yes (vascularization) [74]	No [37]
Elastography	High sensitivity and specificity	Requires specific software and specialized operator	Yes (elasticity)	No [37]
CEUS	Progression of vessel formation [20]	Low sensitivity and specificity [20]	Yes [54]	Yes [54]
Nuclear medicine	Information through true functional imaging [34]	Little anatomical information, low sensitivity, cost, low availability, need for radioactive tracers [34]	Yes [27,61]	Yes [65]
MRI	Differentiation of soft tissues, high resolution [34]	Very high cost, low availability, need for anesthesia [34]	Yes [70]	Yes [71]

**Table 2 animals-15-03506-t002:** Comparative summary of the diagnostic imaging features of benign and malignant mammary gland tumors in the dog and cat (primary lesions).

	Benign MGT	Malignant MGT	Comments
**Size**	Tend to be smaller [40,75,77]	Tend to be larger [75,77]	There is no consensual cut off
Margins/Contour	Usually well-defined, smooth, regular [40,75,76]	Usually ill-defined, irregular, spiculated [76,78,82]	Non-defined margins were not reliable in differentiating benign/malignant lesions [78]
Shape	Round to oval [76,78]	Irregular, microlobulated [78]	Malignant tumors often distort surrounding tissue
Local invasion	Tend to be isolated [40,76,78]	Tend to invade fascia, muscle, skin [76,82]	Best seen on MRI/CT
Echotexture Echogenicity (US)	Homogeneous, hypoechoic or isoechoic [40,75,78]	Heterogeneous, mixed echogenicity [75,76,78]	Heterogenicity due to edema, hemorrhage, calcification, cystic/necrotic areas [75]
Posterior acoustic features (US)	Shadowing or enhancement more likely [78,80]	Shadowing [40] or enhancement more likely [75,76,79]	Features not consensual [78,81]
Vascularity (Color Doppler US)	Limited angiogenesis [75,81]. Peripheral vascular pattern [74,81]	Increased neovascularization [40,87]. Mixed vascular pattern [74,81]	Several studies report no significant differences in vascular flow [75,81,88], but in the distribution pattern [81]
Elastography (strain, shear-wave, ARFI)	Low stiffness and greater deformability [44,83,87]	Lower tissue deformation [40]. Stiffness values exceeding 80–100 kPa [44,83,87]	Malignancy evidence denser stroma matrix and reduced elasticity [85]
CEUS	May have low specificity in differentiating benign/malignant tumors [40]	Shorter periods of contrast wash-in and peak enhancement times in complex carcinomas [40,77]	Could be a useful adjunct tool for assessing tumoral perfusion patterns
Contrast Enhancement (CT/MRI)	Mild, homogeneous [72]	Strong, heterogeneous enhancement [26]	Dynamic contrast-enhanced MRI may show rapid wash-in/wash-out in malignancy
FDG-PET/CT	Glucose uptake < 2 [27]	Glucose uptake > 2 evidenced 100% sensitivity [27]	100% predictive value if lesion > 1.5 cm + glucose uptake > 2 [27]
Cysts	May occur [75]	Cystic degeneration [72]	Proportion of cystic lesions is similar in both [75]
Necrosis	Less common [1]	Common, central necrosis [72]	May help differentiate high-grade malignant tumors [1]
Calcifications	Occasional	May be present, coarse or microcalcifications	Accurate detection by CT [72]

## Data Availability

The original contributions presented in this study are included in the article. Further inquiries can be directed to the corresponding authors.

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
