# Peer review of "Diagnostic Imaging Features of Mammary Gland Tumors in Dogs and Cats"

_animals, 2025, doi:10.3390/ani15243506_

Round 1

Reviewer 1 Report

Comments and Suggestions for Authors

Dear authors,

Some appointments were prepared in order to clarify and improve the article.

  • Line 44: consider chance the key word “mammary gland tumor” because it is presented at title.
  • Line 42: the information “Among these, around 60% are benign and 40% malignant [3].” must consider the local of the study; it is recognized that close to cropped area (as Australia and Brazil), the malignant mammary tumors are more frequent than benign and metastasis are common finds. Consider adjust the information considering the diagnosis observed in different places.
  • Line 68: “masses” is not a synonymous of tumors/neoplasia; consider change this term;
  • Line 105, 114, 149, 152, 210, 228, 243, 254: veterinary medicine is a science, consider use capital letters;
  • Line 338-340: the information “These conflicting findings indicate that acoustic enhancement and shadowing alone are not reliable criteria for distinguishing between benign and malignant mammary tumors” are a conclusion obtained in the article or by the authors?
  • Some of the figures presented in the article appears from another article, is it an adequate interpretation? If yes, the source is not presented in the article. If not, please make clear the image author;
  • Final consideration is not more appropriate than conclusion?

Author Response

Dear reviewer,

We sincerely appreciate the time and effort you have devoted to providing such valuable feedback, which will help us enhance the quality and clarity of our paper.

We have thoroughly addressed all your comments, and we hope our responses and manuscript changes (in blue) meet your expectations.

Cláudia Baptista

Dear authors,

Some appointments were prepared in order to clarify and improve the article.

Line 44: consider change the key word “mammary gland tumor” because it is presented at title.

A: Thank you for your suggestion. We have replaced “mammary gland tumor” with “mammary neoplasia” to broaden the indexing scope.

Line 42: the information “Among these, around 60% are benign and 40% malignant [3].” must consider the local of the study; it is recognized that close to cropped area (as Australia and Brazil), the malignant mammary tumors are more frequent than benign and metastasis are common finds. Consider adjust the information considering the diagnosis observed in different places.

A: Thank you for this valuable observation. We agree that the proportion of benign and malignant mammary tumors in dogs is not uniform across geographical regions. The classical 60/40 distribution refers to global estimates from older literature; however, more recent reviews highlight marked geographic variability influenced by differences in neutering practices, socioeconomic factors, and population management. In order to clarify this point, we have revised the text to acknowledge this heterogeneity. A sentence has been added to the Introduction referring to the review by Vazquez et al. (2023), which reports significant geographic variation in both incidence and malignancy rates of canine mammary tumors. The updated text has been incorporated and highlighted in blue in the revised manuscript.

Line 68: “masses” is not a synonymous of tumors/neoplasia; consider change this term;

A: Thank you for your comment. We have replaced “masses” with “tumors” to improve clarity. This change has been implemented and is highlighted in blue in the revised manuscript.

Line 105, 114, 149, 152, 210, 228, 243, 254: veterinary medicine is a science, consider use capital letters;

A: Thank you for your observation. We have corrected this throughout the manuscript in all indicated lines. The changes have been implemented and are highlighted in blue in the revised version.

Line 338-340: the information “These conflicting findings indicate that acoustic enhancement and shadowing alone are not reliable criteria for distinguishing between benign and malignant mammary tumors” are a conclusion obtained in the article or by the authors?

A: Thank you for your comment. This statement is not a novel conclusion derived from our study; rather, it is an interpretation based on the conflicting results reported in the cited literature. Different authors have described acoustic enhancement and shadowing as associated with malignancy [79], with benignity [80], or occurring equally in both tumor types [75,81]. These inconsistencies collectively indicate that these sonographic features cannot be used reliably as sole criteria for differentiating benign from malignant mammary tumors. To clarify this point, we have modified the sentence to explicitly state that this interpretation is based on previously published studies. The revision has been highlighted in blue in the updated manuscript.

Some of the figures presented in the article appears from another article, is it an adequate interpretation? If yes, the source is not presented in the article. If not, please make clear the image author;

A: Thank you for bringing this to our attention. We would like to clarify that all figures included in the manuscript are original and were produced by the authors using images obtained during our clinical and imaging evaluations. None of the images were taken or reproduced from previously published articles. To avoid any misunderstanding, we have added a clarifying statement in the end of the section introduction.

Final consideration is not more appropriate than conclusion?

A: Thank you for your suggestion. We agree that “Final Considerations” could be an appropriate and clear heading for the closing section of a review article. We have therefore replaced “Conclusion” with “Final Considerations” in the revised manuscript, as recommended.

Reviewer 2 Report

Comments and Suggestions for Authors

Dear Authors,

In my opinion, the manuscript presents a thorough and well-organized review of imaging modalities used in the diagnosis and staging of mammary gland tumors in dogs and cats. The breadth of literature covered and the integration of recent studies significantly enhance the scientific value of the work. The sections are well structured, with logical transitions between imaging techniques. Figures are clear, well chosen, and appropriately support the text. The explanations of modality-specific strengths and limitations are particularly helpful for clinical readers. The manuscript provides solid analytical depth, including comparative sensitivity/specificity data and correlations with histopathology. This strengthens its relevance for both clinicians and researchers. The emphasis on multimodal imaging is well-justified and appropriate; however, histopathology remains the gold standard in diagnostics. I propose considering an expansion of the conclusion with more explicit, practical recommendations for clinicians on when to select each imaging modality. You may also wish to briefly summarize future directions in advanced imaging (e.g., quantitative CEUS, radiomics, AI-assisted diagnostics), as this would add additional value to the review.

Please see the attachment for more comments.

Best regards

Author Response

Dear reviewer,

We sincerely appreciate the time and effort you have devoted to providing such valuable feedback, which will help us enhance the quality and clarity of our paper.

We have thoroughly addressed all your comments, and we hope our responses and manuscript changes (in green) meet your expectations.

Cláudia Baptista

Dear Authors,

In my opinion, the manuscript presents a thorough and well-organized review of imaging modalities used in the diagnosis and staging of mammary gland tumors in dogs and cats. The breadth of literature covered and the integration of recent studies significantly enhance the scientific value of the work. The sections are well structured, with logical transitions between imaging techniques. Figures are clear, well chosen, and appropriately support the text. The explanations of modality-specific strengths and limitations are particularly helpful for clinical readers. The manuscript provides solid analytical depth, including comparative sensitivity/specificity data and correlations with histopathology. This strengthens its relevance for both clinicians and researchers. The emphasis on multimodal imaging is well-justified and appropriate; however, histopathology remains the gold standard in diagnostics. I propose considering an expansion of the conclusion with more explicit, practical recommendations for clinicians on when to select each imaging modality. You may also wish to briefly summarize future directions in advanced imaging (e.g., quantitative CEUS, radiomics, AI-assisted diagnostics), as this would add additional value to the review.

Please see the attachment for more comments.

Best regards

A: We sincerely thank you for your thorough and constructive review of our manuscript. We greatly appreciate your positive feedback regarding the structure, clarity, scientific value, and clinical relevance of the work. Your recognition of the breadth of literature reviewed and the usefulness of the comparative analysis is very encouraging.

We also thank you for your insightful suggestions regarding the expansion of the conclusion, particularly the inclusion of more explicit, practical recommendations for clinicians and a brief summary of future directions in advanced imaging techniques. These recommendations are highly valuable and we have incorporated them into the revised manuscript (green).

Please find below a point-by-point response to all comments. Again, all changes made to the manuscript are highlighted in green for ease of review.

Additional Reviewer Report – Manuscript ID: animals-4004214

  1. Main question addressed by the research

The manuscript investigates the usefulness of various imaging modalities—radiography, ultrasonography, Doppler techniques, CT, CEUS, elastography, MRI, and nuclear medicine—in diagnosing, characterizing, and staging mammary gland tumors in dogs and cats. It aims to consolidate current knowledge and evaluate the diagnostic value and limitations of these techniques in veterinary oncology.

  1. Originality, relevance, and gap addressed

The topic is highly relevant to veterinary diagnostic imaging and comparative oncology. Although imaging of mammary tumors is not new, the manuscript attempts to synthesize recent technological developments (such as elastography, CEUS, and AI-enhanced modalities).

The review addresses an existing gap by:

.providing a broad, method-by-method comparison,

  • .emphasizing emerging imaging modalities,
  • .integrating findings from multiple species (dog and cat) and multiple imaging approaches.

This comparative synthesis is valuable because recent literature tends to treat individual techniques in isolation.

A: Thank you for highlighting the relevance and contribution of our manuscript.

  1. Contribution to the field compared with existing literature

Compared to previously published reviews, this manuscript: includes newer studies from 2018–2024, integrates quantitative performance indicators (e.g., sensitivity/specificity of CT, Doppler, elastography), links imaging findings to histopathological subtypes, discusses multimodal imaging strategies for better staging and treatment planning.

These aspects make the review more comprehensive and clinically useful.

A: Thank you for these positive remarks. We appreciate that this multimodal approach was considered comprehensive and clinically useful.

  1. Methodology – suggested improvements

Although it is a review article, several methodological aspects of the literature selection and synthesis could be strengthened:

4.1. Literature search description

The manuscript does not clearly describe:

  • search terms used,
  • databases consulted,
  • inclusion and exclusion criteria,
  • date ranges,
  • methodology for evaluating study quality.
  • Adding a transparent search strategy would significantly improve rigor.

A: Thank you for this suggestion. Although this manuscript is a narrative (non-systematic) review, we agree that providing additional transparency regarding the literature search improves clarity and rigor. We have therefore added a brief description of the search strategy, including the databases consulted, general search terms, time frame, and basic inclusion criteria. This information has been incorporated into the revised manuscript and highlighted in green.

4.2. Critical evaluation vs. descriptive synthesis

In my opinion, some sections summarize information without critically evaluating:

  • limitations of available studies,
  • variability in diagnostic accuracy across studies,
  • heterogeneity in sample populations,
  • differences in protocols and machine parameters.

I believe that a more rigorous critical appraisal would enhance the scientific value.

A: We appreciate this observation. Our intention was to provide a balanced synthesis of the available studies; however, we agree that emphasising study limitations and sources of heterogeneity strengthens the review. We have therefore expanded the critical appraisal across several sections, including comments regarding sample size variability, differences in imaging protocols and equipment, and the resulting variability in diagnostic accuracy reported in the literature. These additions are now highlighted in green.

4.3. Clinical applicability

The review would benefit from:

- more explicit guidance on when each imaging technique should be chosen,

- practical diagnostic algorithms or flowcharts,

- Differentiation between techniques appropriate for staging vs. screening.

A: Thank you for this valuable suggestion. We have expanded the Final Considerations to include clearer, modality-specific recommendations for clinicians, differentiating techniques more suitable for screening from those used for staging. Given the narrative nature of the review, we did not include a formal diagnostic algorithm, but we have added concise practical guidance to support clinical decision-making and a new table (Table 2). These additions are highlighted in green.

4.4. Species differences

Cats and dogs are sometimes discussed interchangeably; highlighting differences in tumor behavior and imaging reliability between species would add scientific nuance.

A: We agree that highlighting species-specific considerations adds scientific nuance. We have therefore clarified, where relevant, differences in tumor behavior, prevalence, and the reliability of imaging modalities between dogs and cats. These clarifications are included in the revised manuscript and highlighted in green.

  1. Evaluation of conclusions

The conclusions are generally consistent with the literature presented, but they could be improved by:

  • explicitly restating the main question and answering it directly,
  • presenting clear, actionable recommendations for clinicians – in my opinion, significant for practitioners
  • discussing limitations in current evidence and technological access,
  • avoiding overly general statements (e.g., “imaging is essential” without specifying which modality in which clinical context).

Overall, the conclusions are acceptable but would benefit from greater synthesis and specificity.

A: Thank you for these valuable suggestions. In response, we have revised the Final Considerations to improve clarity and specificity. We have explicitly restated the central aim of the review and provided more direct, actionable recommendations for clinicians regarding the selection and use of each imaging modality. We have also added a brief discussion of the main limitations in the current evidence, including variability in study design, protocol heterogeneity, and differences in equipment availability. General statements have been refined to ensure that each conclusion is tied to a specific clinical context. All modifications are highlighted in green in the revised manuscript.

  1. Tables and Figures

The figures are high-quality and visually support the text well.

Some legends could be expanded with more detailed explanations (e.g., modality parameters or diagnostic significance).

If possible, provide more schematic diagrams rather than only imaging examples—for example, illustrative color-coded elastography maps or diagnostic decision trees.

A: Thank you for your positive assessment of the figures. We appreciate your suggestion to expand some of the figure legends. In response, we have revised several legends to provide clearer descriptions of modality parameters and their diagnostic significance. These changes are highlighted in green in the revised manuscript.

In order to provide more schematic information, we introduced in the paper a second table that summarises differential imaging features of MGT primary tumors (benign versus malignant). We hope that this solution will meet your expectations.

Reviewer 3 Report

Comments and Suggestions for Authors

General Comment:
Although the topic is not new, the authors present an interesting and valuable review of the diagnostic imaging features of mammary gland tumors in dogs and cats. The manuscript fits well within the scope of the journal, and the writing style is appropriate.

Title:
The title is short, concise, and adequate. 

Simple Summary:
The full name of each abbreviation should be provided before its first use (e.g., computed tomography (CT), magnetic resonance imaging (MRI)).

Abstract and Keywords:
The abstract is complete. However, the keywords should differ from those used in the title.

2. Imaging Techniques for the Diagnosis and Evaluation of Disease Progression

Once defined, abbreviations should be used consistently throughout the text (e.g., “computed tomography” (line 145), “ultrasonography” (line 161), “positron emission tomography” (line 215)).

The Power Doppler and B-Flow modalities available in some US systems should be addressed in this section.

The most commonly used contrast agents should also be described. Additionally, the types of evaluations possible with CEUS - both qualitative and quantitative, including time-intensity curve (TIC) analysis - should be discussed.

3. Imaging Features of Primary Lesions

Figure 1A should be replaced; the rectangle under the mammary tumor should be removed.

Images obtained using Doppler mode should be included.

MRI images should also be added.

It should be clarified whether owners’ consent was obtained for the use of the images.

4. Imaging Features of Metastatic Lesions

The methods should be presented in the same order as in Section 3 to ensure consistency.

Conclusion:
The conclusions are adequate.

References:
The references are appropriate.

Author Response

Dear reviewer,

We sincerely appreciate the time and effort you have devoted to providing such valuable feedback, which will help us enhance the quality and clarity of our paper.

We have thoroughly addressed all your comments, and we hope our responses and manuscript changes (in magenta) meet your expectations.

Cláudia Baptista

General Comment:

Although the topic is not new, the authors present an interesting and valuable review of the diagnostic imaging features of mammary gland tumors in dogs and cats. The manuscript fits well within the scope of the journal, and the writing style is appropriate.

A: We are sincerely grateful for the positive overall assessment of our manuscript.

Title:

The title is short, concise, and adequate.

A: Thank you.

Simple Summary:

The full name of each abbreviation should be provided before its first use (e.g., computed tomography (CT), magnetic resonance imaging (MRI)).

A: We thank you for this helpful observation. In the revised version, we have ensured that all abbreviations appearing in the Simple Summary are written out in full upon first mention (e.g., computed tomography (CT), magnetic resonance imaging (MRI)). These changes have been incorporated and are highlighted in magenta in the updated manuscript.

Abstract and Keywords:

The abstract is complete. However, the keywords should differ from those used in the title.

A: We thank you for this comment. The keywords have been revised to ensure that they do not duplicate terms used in the title.

  1. Imaging Techniques for the Diagnosis and Evaluation of Disease Progression

Once defined, abbreviations should be used consistently throughout the text (e.g., “computed tomography” (line 145), “ultrasonography” (line 161), “positron emission tomography” (line 215)).

The Power Doppler and B-Flow modalities available in some US systems should be addressed in this section.

The most commonly used contrast agents should also be described. Additionally, the types of evaluations possible with CEUS - both qualitative and quantitative, including time-intensity curve (TIC) analysis - should be discussed.

A: We thank you for this comment. All abbreviations (e.g., ultrasonography (US), computed tomography (CT), positron emission tomography (PET)) have now been written out in full upon first mention and used consistently throughout the manuscript. These corrections have been implemented and appear highlighted in magenta in the revised version. We have added a short paragraph in the section on ultrasonography describing Power Doppler and the relevance in the evaluation of vascular patterns in mammary gland tumors. Also, a new subsection has been added to briefly describe the most used contrast agents in veterinary CEUS and to clarify the possible qualitative and quantitative assessments, including time–intensity curve (TIC) analysis. These additions are now incorporated in Section 2 and highlighted in magenta.

  1. Imaging Features of Primary Lesions

Figure 1A should be replaced; the rectangle under the mammary tumor should be removed.

Images obtained using Doppler mode should be included.

MRI images should also be added.

It should be clarified whether owners’ consent was obtained for the use of the images.

A: Regarding Figure 1A, the rectangle under the mammary tumor was removed.

As stated in the manuscript (at the end of the Introduction, in blue), all figures are original imaging studies acquired by the authors. We apologize that we do not have informative, high-quality Doppler or MRI images of MGTs available. However, to compensate for this limitation, we have added a second table that summarizes the key differential imaging features of primary MGT tumors. This allows readers to easily compare and understand relevant characteristics related to Doppler and MRI modalities (and others), even without the corresponding figures. We hope that this solution satisfactorily addresses your concerns.

Finally, we confirm that owners’ consent was obtained prior to all imaging procedures. As our hospital is a teaching and referral institution, tutors routinely sign an informed consent document at admission authorizing the use of anonymized clinical information and imaging data for research and educational purposes. This information has now been added at the end of the manuscript “Informed Consent Statement”.

  1. Imaging Features of Metastatic Lesions

The methods should be presented in the same order as in Section 3 to ensure consistency.

A: We thank you for this thoughtful observation. We agree that consistency in the structure of scientific manuscripts is generally important. However, in the present review, we intentionally organized Section 2 (Imaging Techniques) according to the logical progression of diagnostic imaging principles—starting from the most widely available, first-line modalities and moving towards advanced or specialized techniques. Section 3, on the other hand, is structured based on clinical presentation and staging needs, which naturally led to a different sequence of modalities. Because each section serves a different purpose within the manuscript, mirroring the order between them would reduce clarity rather than enhance it. For this reason, and to preserve the readability and pedagogical flow of the review, we have opted to maintain the current structure. We hope you can agree that this approach ultimately supports a more coherent understanding of the diagnostic pathway. Nonetheless, all sections have been carefully reviewed to ensure internal consistency and smooth cross-referencing.

Conclusion:

The conclusions are adequate.

References:

The references are appropriate.

A: We thank you for the positive feedback regarding both the conclusions and the reference list.

Round 2

Reviewer 3 Report

Comments and Suggestions for Authors

Accept in present form.